# Targeting Multiple Homeostasis-Maintaining Systems by Ionophore Nigericin Is a Novel Approach for Senolysis

**DOI:** 10.3390/ijms232214251

**Published:** 2022-11-17

**Authors:** Pavel I. Deryabin, Alla N. Shatrova, Aleksandra V. Borodkina

**Affiliations:** 1Mechanisms of Cellular Senescence Group, Institute of Cytology of the Russian Academy of Sciences, Tikhoretsky Avenue 4, 194064 Saint-Petersburg, Russia; 2Laboratory of Intracellular Membranes Dynamic, Institute of Cytology of the Russian Academy of Sciences, Tikhoretsky Avenue 4, 194064 Saint-Petersburg, Russia

**Keywords:** senescence, nigericin, senolysis, pyroptosis, FOXO4-DRI, dasatinib + quercetin

## Abstract

Within the present study we proposed a novel approach for senolysis based on the simultaneous disturbance of the several homeostasis-maintaining systems in senescent cells including intracellular ionic balance, energy production and intracellular utilization of damaged products. Of note, we could not induce senolysis by applying ouabain, amiloride, valinomycin or NH_4_Cl—compounds that modify each of these systems solely. However, we found that ionophore nigericin can disturb plasma membrane potential, intracellular pH, mitochondrial membrane potential and autophagy at once. By affecting all of the tested homeostasis-maintaining systems, nigericin induced senolytic action towards stromal and epithelial senescent cells of different origins. Moreover, the senolytic effect of nigericin was independent of the senescence-inducing stimuli. We uncovered that K^+^ efflux caused by nigericin initiated pyroptosis in senescent cells. According to our data, the higher sensitivity of senescent cells compared to the control ones towards nigericin-induced death was partially mediated by the lower intracellular K^+^ content in senescent cells and by their predisposition towards pyroptosis. Finally, we proposed an interval dosing strategy to minimize the negative effects of nigericin on the control cells and to achieve maximal senolytic effect. Hence, our data suggest ionophore nigericin as a new senotherapeutic compound for testing against age-related diseases.

## 1. Introduction

Currently, senescence is a well-established phenomenon, which, in relation to proliferating cells, is defined as the irreversible loss of proliferation of metabolically active cells resulting predominantly from DNA damage [1]. Senescence directly mediates loss of cellular and tissue functionality and thus is largely associated with the progression of various pathologies, including osteoarthritis, fibrotic pulmonary disease, hepatic steatosis, neurodegeneration, atherosclerosis, cardiovascular disease, diabetes, and kidney failure [2,3,4,5,6,7,8,9,10,11,12,13,14]. In our recent study we also demonstrated that premature senescence of endometrial stromal cells might impair endometrial stroma functioning and disturb embryo implantation [15]. In light of these findings, the possibility for the targeted elimination of senescent cells seems to be a rather promising strategy for treating diseases of different etiologies. Indeed, the development of senolytics—compounds capable of targeted killing of senescent cells—is one of the core trends in the biology of aging [1,16]. To date, impressive results based on senolytics application are obtained for treating diseases in a wide range of disease models [1,2,3,4,5,6,7,8,9,10,11,12,13,14]. Moreover, clinical trials on senolytics application are already underway or about to begin [10,17,18,19,20]. Nevertheless, it is now clear that senolytic drugs have significant limitations in terms of specificity and broad-spectrum activity. Varying sensitivity of different types of senescent cells towards senolytics might be due to the heterogeneity in senescence programs [2,21,22,23].

For now, the following approaches to search for senolytics have already been proposed: (1) targeting anti-apoptotic/pro-survival networks in senescent cells (BCL-2 family members, USP7, HSP90, etc.); (2) targeting altered organelles in senescent cells (reducing the abundance, enzymatic activity and permeability of mitochondria or lysosomes); (3) targeting senescent cells ‘antigens’ (surface markers, receptors); and (4) modifying separate biochemical changes (pH, buffer systems, membrane potential) reviewed in [16]. Within the present study we suggested the multi-parametric approach based on the complex biochemical changes to uncover the potential senolytic compounds. The underlying idea of this approach can be simply explained by the following analogy: senescent cells can be considered as a badly damaged car (reflecting improper functioning and intracellular damages in senescent cells) that retains the ability to move in emergency mode (reflecting preserved viability and metabolic activity). From this point of view, a small obstacle on the road that can be easily overcome (compensated) by a normally operating car (normal cells) because of the proper functioning of the damper systems (homeostatic and compensatory systems), should become critical for a damaged car (senescent cells). To create the so-called ‘obstacle on the road’ here, we altered the functioning of the main homeostasis maintaining systems, including intracellular ionic balance, energy production and intracellular utilization of damaged products. Interestingly, we could not induce senolysis by modifying each of these parameters solely. However, combined action on these homeostasis-maintaining systems by ionophore nigericin was sufficient to induce senolysis. Moreover, since the proposed approach is based on the disturbance of the several homeostasis-maintaining systems, the senolytic effect of nigericin was independent of the cell type and senescence-inducing stimuli. Therefore, the proposed approach to search for the compound with senolytic properties allowed us to uncover a novel broad-spectrum senolytic—nigericin—and to reveal the molecular mechanisms of its action.

## 2. Results

### 2.1. Senescent Endometrial Stromal Cells (End-Mscs) Display Resistance towards Various Senolytic Compounds

In our recent study we revealed that cardiac glycosides, considered as the broad spectrum senolytic compounds, cannot induce senolysis in human mesenchymal stromal cells of various origin, including adipose tissue, dental pulp, Warton jelly, and endometrium [23]. Here we extended the list of senolytics and tested for the effects of FOXO4-DRI, quercetin, and the combination of dasatinib plus quercetin using the model of stress-induced senescence of END-MSCs. These compounds are stated to induce senolysis via different mechanisms: FOXO4-DRI is a small penetrating peptide that perturbs the FOXO4 interaction with p53, resulting in p53 nuclear exclusion and cell-intrinsic apoptosis; quercetin is a natural flavonol that inhibits PI3K, other kinases, and serpines; dasatinib is the Src kinase inhibitor [2,24]. To trigger senescence, END-MSCs were treated with sublethal oxidative stress, and the detailed description of senescence-inducing conditions is provided in the Materials and Methods section. The main features of senescent END-MSCs are presented at Figure 1a–g. Notably, neither FOXO4-DRI nor quercetin or the combined action of dasatinib plus quercetin could induce death selectively in senescent END-MSCs, indicating the absence of senolysis (Figure 1H).

### 2.2. Adaptation to Senescent State Is Accompanied by the Altered Homeostatic Properties of the END-MSCs

The inability of the applied compounds to induce senolysis prompted us to search for another approach to ‘kill’ senescent END-MSCs. Bearing in mind the idea of the senescent cell as ‘a badly damaged car’, we further tried to answer how senescent END-MSCs adopted their ‘damper systems’ to these ‘damages’ to preserve viability.

The cell membrane forms the physical barrier responsible for the maintenance of the intracellular homeostasis due to its selective permeability for various macromolecules, water and ions. In non-excitable cells the membrane potential is considered to be a key biophysical signal that regulates important activities, including cell cycle progression, proliferation, migration and differentiation [25]. Thus, we speculated that one of the adaptation mechanisms in senescent END-MSCs could be realized via altered membrane properties. Indeed, using fluorescent probe DiBAC4 (3), we revealed that senescent END-MSCs displayed increased fluorescence of the dye, indicating the more depolarized membrane compared to the control cells (Figure 2A). Plasma membrane potential (PMP) reflecting electrical potential difference between the cell cytoplasm and the extracellular environment is created via the multiple ion channels mediating controlled and directed transport of specific ions through the membrane [25]. Indeed, when we annotated differentially expressed genes between senescent and control END-MSCs in GO terms, we revealed the activation of the processes related to the regulation of cation transmembrane transport and the regulation of membrane potential in senescent cells (Figure 2B).

Permanently altered ion transport through the membrane should inevitably lead to disturbed intracellular ion homeostasis, particularly to modified intracellular pH (pH_i_). Thus, we next evaluated whether senescent END-MSCs undergo intracellular (pH_i_) alterations. To do so, we used specific fluorescent pH-indicator BCECF-AM. As shown in Figure 2C, pH_i_ was significantly lower in senescent END-MSCs compared to the control ones. In line with this result, a bioinformatic analysis revealed the upregulation of the processes related to pH regulation and response in senescent cells (Figure 2D). The identified GO terms describe processes involved in the maintenance of an internal equilibrium of hydrogen ions, thereby modulating the internal pH_i_ within the cell.

The acidification of cytosol should -cause decreased pH in the intermembrane space of the mitochondrion, since H^+^ ions equilibrate freely across its outer membrane [26]. The latter would change the proton gradient between the intermembrane space and the mitochondrial matrix, and thus would affect the proper functioning of the electron transport chain and ATP production [26]. Indeed, we revealed decreased mitochondrial membrane potential (MMP) in senescent END-MSCs compared to control cells (Figure 2E). Moreover, we detected the downregulation of the processes related to mitochondrial transmembrane transport and mitochondrial ATP synthesis coupled electron transport in senescent END-MSCs (Figure 2F). These data suggest the mitochondrial dysfunction in senescent cells.

To cope with the decreased ATP generation, cells should shift energy metabolism from energy-consuming synthetic processes towards catabolic ones [27]. The activation of AMP-activated protein kinase (AMPK) marks this shift and triggers autophagy—a conserved catabolic process that degrades cytoplasmic constituents and organelles in the lysosomes, thus compensating for the energy loss. In line with this notion, senescent END-MSCs displayed the activation of AMPK, ULK, the enhanced expression of Atg9 and the accumulation of the lipidated isoform of LC3 (Figure 2H). These results were additionally verified by the functional annotation of the differentially expressed genes between senescent and control cells (Figure 2I). Besides energy production, autophagy is required to maintain intracellular homeostasis by degrading damaged organelles and macromolecules. Despite the increased activity of the main protein components of the autophagic machinery, senescent END-MSCs characterized by the accumulation of lipofucine granules composed of damaged macromolecules as indicated by the significantly elevated autofluorescence. This suggests either for the insufficient autophagy progression or for the significant overload of this system (Figure 2G).

Together, the data presented above demonstrate that most of the homeostasis-maintaining systems are disturbed and overloaded in senescent cells, thus it seems reasonable to affect these systems to ‘kill’ senescent END-MSCs.

### 2.3. Imbalancing Each ‘Damper’ System Is Insufficient to Trigger Senolysis in END-MSCs

Having established the precise alterations of the tested homeostasis-maintaining systems in senescent END-MSCs, we further applied compounds to imbalance them. The idea was that the homeostasis-maintaining systems in senescent cells should be much closer to their tensile strength than in the control ones. Thus, a lesser impact could be sufficient to imbalance the systems and to induce death selectively in senescent END-MSCs.

In line with this hypothesis, we recently tested for the senolytic activity of ouabain towards senescent END-MSCs [23]. Ouabain is a widely used inhibitor of Na^+^/K^+^ ATPase, which regulates membrane potential by maintaining sodium and potassium ion distribution. As expected, the treatment of control and senescent END-MSCs with ouabain resulted in significant depolarization of the plasma membrane (Figure 3A). Importantly, the fluorescence of DiBAC4 (3) dye in control ouabain-treated END-MSCs only reached the basal fluorescence level of that in senescent END-MSCs, while upon ouabain treatment, fluorescence of the dye in senescent cells enhanced far more significantly (Figure 3A). Despite such a dramatic alteration in membrane potential caused by ouabain, senescent cells could cope with it, since this compound could not trigger selective death in senescent END-MSCs (Figure 3B).

We further speculated that more pronounced acidification of cytosol could trigger senolysis in END-MSCs. To decrease intracellular pH levels, we applied amiloride, a potent inhibitor of Na^+^/H^+^ exchange through the plasma membrane, which ensures the outflow of hydrogen ions from the cell. Indeed, amiloride treatment led to a significant decrease of pH_i_ both in control and in senescent cells, however it did not result in senolysis (Figure 3c,d).

The next point was to cause MMP disruption. To this end, we applied valinomycin. This compound forms lipid-soluble complexes with K^+^ that can easily penetrate into the mitochondrial membrane. The matrix side of the inner mitochondrial membrane has a negative potential and positively charged valiomycin-K^+^ complexes are drawn inward, thus decreasing MMP. As shown in Figure 3E, valinomycin application led to a pronounced MMP drop both in control and senescent END-MSCs, though it did not affect viability of the cells (Figure 3F).

Finally, we tried to worsen autophagy in senescent END-MSCs by applying NH_4_Cl—a well-known inhibitor of autophagy. Upon NH_4_Cl treatment, we detected the dramatic elevation of autofluorescence and the pronounced accumulation of the lipidated isoform of LC3, both indicating for the block of autophagy in control and senescent END-MSCs (Figure 3G,H). However, the treatment with NH_4_Cl was insufficient to trigger selective death in senescent END-MSCs, demonstrating the absence of senolysis (Figure 3I).

### 2.4. Nigericin Triggered Senolysis in END-MSCs Via Complex Action on The ‘Damper’ Systems of Senescent Cells

The data above demonstrates that disturbing each ‘damper’ system solely is insufficient to trigger senolysis in senescent END-MSCs. Therefore, we searched for the approach to perform complex action on these systems. Based on the literary findings, we focused on nigericin—a lipid-soluble antibiotic that mediates the electrically neutral exchange of H^+^ for K^+^ [28]. Being a mobile ionophore, nigericin forms complexes with K^+^ at one side of the membrane and then migrates to the other side, where K^+^ is discharged and the anionic form is protonated. The moving force for this exchange is the concentration gradient for K^+^ or H^+^. Intracellular concentrations of K^+^ are much higher than those in extracellular space, thus nigericin would cause the leak of K^+^ ions from the cell and the simultaneous entrance of H^+^. As a result of such an exchange, the level of intracellular K^+^ should decrease and disturb PMP, while intracellular H^+^ should increase and cause intracellular acidification. Indeed, we revealed that treatment with nigericin led to the increased fluorescence of DiBAC4 (3) dye, reflecting membrane depolarization in control and senescent END-MSCs (Figure 4A). At the same time, the application of nigericin resulted in lowered pH_i_ in control and senescent END-MSCs (Figure 4B).

Nigericin can rapidly diffuse through mitochondrial membranes and facilitate the exchange of K^+^ with H^+^ between the mitochondrial matrix and intermembrane space [29]. Normally, the mitochondrial matrix pH is higher than the pH of the intermembrane space because of the expulsion of protons across the inner membrane by the components of the electron transport chain. This transmembrane pH gradient generates a proton motive force, which is coupled with the membrane potential of mitochondria to drive ATP synthesis. Nigericin allows H^+^ to pass across the membrane in the direction of the concentration gradient (i.e., inward mitochondria), thereby disrupting the mitochondrial proton gradient and uncoupling oxidative phosphorylation. Interestingly, we observed an increase in MMP both in control and senescent END-MSCs 3 days after nigericin application (Figure 4C).

Finally, we tested whether nigericin would affect autophagy in END-MSCs. Since nigericin acts as an antiporter of K^+^/H^+^, it is shown to raise pH of the acidic compartments, leading to reduced lysosomal protein degradation and inhibiting fusion between autophagosomes and lysosomes [30]. The application of nigericin resulted in enhanced phosphorylation of AMPK along with the accumulation of LC3-II (Figure 4E). At the same time, we revealed a significant increase in autofluorescence, indicating the lipofuscine accumulation (Figure 4D). Thus, on the one hand, nigericin enhanced autophagosome formation and, on the other, led to their accumulation, impairing proper autophagy progression.

Based on the facts above, nigericin performs a complex action on the ‘damper’ systems of cells, what according to our analogy of a senescent cell as ‘a badly damaged car’ should inevitably cause senolysis. Indeed, we revealed that nigericin induced death predominantly in senescent END-MSCs (Figure 4F). As shown in Figure 4F, control cells continued to proliferate upon 1 μM of nigericin, while only 40% of senescent END-MSCs preserved viability; upon 10 μM-over 50% of control cells and only around 10% of senescent cells remained viable. These results indicate that nigericin can be considered as the effective senolytic to at least eliminate senescent END-MSCs.

### 2.5. K^+^ Efflux Caused by Nigericin Initiates Pyroptosis in Senescent END-MSCs

As stated above, nigericin can catalyse electroneutral exchange of K^+^ for H^+^ along the concentration gradient, which would result in a net K^+^ efflux from the cells due to high intracellular K^+^ concentration [28]. We first tested whether the more pronounced death of senescent END-MSCs compared to the control ones might be related to the lower intracellular K^+^ levels. If the latter is true then the application of nigericin would cause a more rapid drop of intracellular K^+^ to the “threshold” levels than in control cells. Firstly, we supplemented culturing media with KCl, which should decrease to some extent the concentration gradient for K^+^ between intra- and extracellular space, thus alleviating the effects of nigericin. Indeed, we observed that the addition of KCl partially rescued senescent END-MSCs from nigericin-induced death (Figure 5A). Importantly, the protecting effect of KCl was more pronounced in senescent END-MSCs than in the control cells.

We next checked the sensitivity of cells with various intracellular K^+^ levels towards nigericin. According to our previous data, the level of intracellular K^+^ relies significantly on the phase of the cell cycle [31]. Namely, the intracellular K^+^ level is low in G0/G1-phase cells and high in S-phase cells. Senescent END-MSCs are blocked in the G0/G1 phase of the cell cycle, and thus should have lower intracellular K^+^ compared to proliferating control cells and also should be more sensitive towards nigericin. To test for this suggestion, control END-MSCs were either blocked in the G0/G1-phase by 18 h serum starvation, or pushed to S-phase by the subsequent addition of the serum-containing media. As shown at Figure 5B, over 90 % of cells were in the G0/G1-phase under starvation conditions (G0/G1-cells), and approximately 30 % of cells reached the S phase 18 h after full media addition (S-cells) (Figure 5B). Importantly, both G0/G1- and S-cells preserved viability. In line with our suggestion, the percent of viable cells remained after nigericin treatment correlated well with the level of intracellular K^+^, i.e., S-cells continued to proliferate, while less than 20 % of G0/G1-cells preserving viability upon nigericin application (Figure 5B). Interestingly, non-synchronized cells were somewhere in the middle, since about 60 % of cells remained viable (Figure 5B). Based on these findings, we can speculate that the more pronounced death-inducing effects of nigericin on senescent cells might be mediated by the more rapid depletion of intracellular K^+^ because of its lower content.

K^+^ efflux acts as an NLRP3 inflammasome activator and pyroptosis inducer [32]. Pyroptosis is type of cell death accompanied by the activation of inflammasomes, pro-inflammatory caspases such as caspase-1, and the maturation of pro-inflammatory cytokines such as interleukin-1β (IL-1β). Lipopolysaccharide is a well-known priming signal for NLRP3 inflammosome activation and pyroptosis [33]. Interestingly, we detected significant enrichment in the lipopolysaccharide-mediated signaling pathway in senescent END-MSCs, which suggested some predisposition of senescent cells towards pyroptosis (Figure 5C). Morphologically, pyroptosis is indicated by cell swelling and plasma membrane rupture, leading to the release of the pro-inflammatory cytokine IL-1β and cellular contents into the extracellular space [32]. In favor of pyroptosis induction, we revealed the enhanced secretion of IL-1β by both control and senescent cells upon nigericin treatment (Figure 5D). Importantly, the level of secreted IL-1β was almost twice higher in senescent END-MSCs than in control cells, suggesting for the more pronounced death of senescent ones (Figure 5D).

Since caspase-1 is the key molecule regulating pyroptosis, we applied Z-VAD-fmk—cell-permeable pan-caspase inhibitor to block its activity. Inhibiting caspase activity partially protected END-MSCs from nigericin-induced death, and the protecting effect was more noticeable in senescent cells (Figure 5F).

Another feature of pyroptosis that is shared with apoptosis is positive staining for annexin V. Contrarily to apoptosis during pyroptosis, annexin V enters the cell via pores and stains the inner side of the membrane. As shown in Figure 5E, the fraction of double positive Annexin V/DAPI cells increased dramatically in senescent cells upon nigericin treatment (Figure 5E).

Based on the data presented above, we can conclude that the application of nigericin results in K^+^ efflux and subsequent pyroptosis induction. More pronounced effects of nigericin on senescent END-MSCs might be related to lower intracellular K^+^ content and predisposition of the latter towards pyroptosis.

### 2.6. Nigericin Has Senolytic Action towards Various Cell Types

In order to extend our observations, we tested for the senolytic action of nigericin using various cell types and senescence models. Specifically, to induce senescence, human mesenchymal stromal cells isolated from dental pulp (DP-MSCs) and Wharton’s jelly (WJ-MSCs) were treated with sublethal doses of doxorubicine (Figure 6A) or oxidative stress (Figure 6C), respectively. A549 lung carcinoma cells were treated with etoposide (Figure 6E). Additionally, we applied the model of replicative senescence for END-MSCs (Figure 6G). Of note, the validity of these senescence models was clearly demonstrated in our recent study [23]. Despite the different dynamics, nigericin induced senolysis in all senescence models and cell types, supporting its broad-spectrum senolytic action (Figure 6B,D,F,H).

### 2.7. Repeated Pulse Treatments with Low Doses of Nigericin Are Effective to Induce Senolysis in Senescent END-MSCs and Are Safe for Control Cells

The in vitro characterization of senolytics is often limited to the verification of selective death induction in senescent cells without paying much attention to the properties of control cells that could also be affected by the compounds. Though nigericin demonstrated clear senolytic properties, it negatively affected the proliferation of the control END-MSCs even at a low concentration of 1 μM (Figure 4F and Figure 6H). At a higher concentration 10 μM, nigericin caused the death of a portion of the control cells (Figure 4F and Figure 6H). To overcome this obstacle, we performed 1 h pulse treatments with 1 μM nigericin repeated weekly. Such a treatment design allowed for the elimination of 90% of senescent cells without dramatic effects on the proliferation of control cells (Figure 7). Therefore, we were able to design an effective and safe scheme to perform nigericin-induced senolysis.

## 3. Discussion

The present study aimed to develop a novel approach to search for the compound with senolytic properties. To this end, we focused on the several biochemical parameters reflecting the altered functioning of homeostasis-maintaining systems in senescent END-MSCs, including PMP, intracellular pH, MMP, and autophagy. Notably, alterations in most of these biochemical properties have been previously examined in the context of cell senescence. For example, various alterations in mitochondrial functioning, including reduced ATP generation, accelerated ROS production and decreased MMP are described as hallmarks of senescent cells [34]. Impaired autophagic flux and lysosomal dysfunction were also found in senescent cells [35,36]. Moreover, partially depolarized plasma membrane and acidified cytosol were recently proven to accompany cell senescence [36,37]. In line with the above, senescent END-MSCs are characterized by the more depolarized plasma membrane, acidified cytosol, decreased MMP and impaired autophagy compared to their young counterparts. Notably, disturbing each of these parameters by the specific compounds—ouabain, amoloride, valinomycin or NH_4_Cl—did not trigger senolysis in END-MSCs. Contrarily, senescent A549, BJ, and IMR90 were found to be sensitive to ouabain treatment [23,37,38]. Furthermore, BRAF-V600E-induced senescent cells turned out to be sensitive for senolysis induced by chloroquine (a compound targeting autophagy) [39]. Based on these data, we can speculate that some types of senescent cells rely more on the proper functioning of the concrete system, thus for these cells, disturbing one parameter might be sufficient to trigger senolysis. However, the absence of senolysis in END-MSCs highlights that such an approach is not universal. Hence, we proposed that complex action on various aspects of senescent cells rather than targeting one sole feature might be a better strategy for directed senolysis.

When looking for the compound to disturb PMP, intracellular pH_i_, MMP and autophagy at once, we paid specific attention to nigericin—an antibiotic derived from the *Streptomyces peucetius* [40]. Nigericin is an H^+^, K^+^ ionophore that can form complexes with K^+^ and transport K^+^ across the lipid bilayer, leading to a decrease in intracellular K^+^ concentration and an increase in H^+^ concentration [40,41,42]. Indeed, we observed that the application of nigericin resulted in plasma membrane depolarization and intracellular acidification in control and senescent END-MSCs. Furthermore, nigericin was previously shown to eliminate the pH gradient across the mitochondrial membrane, leading to a compensatory rise in MMP [43,44]. In line with this notion, we revealed increased MMP in control and senescent END-MSCs upon nigericin. Autophagy induction is considered as an additional mechanism of nigericin action [45,46,47]. For example, in human patient derived glioma cells, nigericin induced AMPK phosphorylation and autophagy [45]. In neuronal cells, nigericin lead to an increase of LC3-II and induced the impairment of autophagic flux [47]. We also detected that the application of nigericin resulted in enhanced phosphorylation of AMPK, the accumulation of LC3-II, and the simultaneous increase in autofluorescence in control and senescent END-MSCs, suggesting the enhanced autophagosome formation and improper autophagy progression. Thus, nigericin influenced all the selected biochemical parameters. Despite the fact that its effects on control and senescent END-MSCs were comparable, we suggested that in senescent cells all the homeostasis-maintaining systems should be much closer to their tensile strength and unbalancing them by nigericin should lead to more pronounced effects on the viability of senescent cells. Indeed, nigericin induced preferential death in senescent END-MSCs, indicating senolysis. Moreover, we verified the broad-spectrum senolytic effect of nigericin by using different cell types and senescence models. This suggests that the complex action on various aspects of senescent cells might be a universal approach.

We further tried to get deeper insight into the molecular mechanisms underlying the higher sensitivity of senescent END-MSCs compared to the control ones towards nigericin-induced death. According to our previous findings, cells blocked in the G0/G1 phase are characterized by significantly lower K^+^ content than proliferating cells [31]. Therefore, the application of nigericin should lead to a more rapid depletion of intracellular K^+^ and death in arrested cells. By synchronizing END-MSCs in various cell cycle phases, we confirmed this suggestion. In line with these results, senescent END-MSCs blocked in the G0/G1 phase should be more sensitive towards the nigericin-induced depletion of intracellular K^+^. The decrease in intracellular K^+^ caused by nigericin is known to activate the NLRP3 inflammasome and to induce pyroptosis [32]. The primary target cells of NLRP3 inflammasome activation and pyroptotic cell death are believed to be immune cells, such as monocytes, macrophages, dendritic cells, and neutrophils [48,49]. However, recent findings reported that other cell types, including epithelial cells and MSCs of various origins, are also prone to inflammosome activation and pyroptosis [50,51,52,53]. By using bioinformatics, we revealed the predisposition of senescent END-MSCs towards pyroptosis. In line with that, senescent END-MSCs secreted increased amounts of IL-1β upon nigericin. Moreover, blocking caspases with Z-VAD-fmk resulted in a pronounced protective effect against pyroptotic death induced by nigericin. Together, these data demonstrate that the senolytic effect of nigericin is partially mediated by the lower intracellular K^+^ content in senescent END-MSCs and by their predisposition towards pyroptosis.

Although senescent END-MSCs turned out to be far more sensitive towards nigericin, this compound affected the properties of the control cells. In fact, all senolytic strategies may elicit undesirable off-target effects, which can be limited by various dosing strategies [16]. According to the literary data, an interval dosing strategy might be rather effective, since senescent cells take 7 days or more to accumulate and develop the senescence-associated secretory phenotype (SASP), at least in vitro, and might potentially take the same time to start re-accumulating in vivo [16,54,55]. Indeed, we were able to minimize the negative effects of nigericin on the control cells, preserving its maximal senolytic effect by performing three rounds of weekly pulse treatments. The possibility of applying nigericin in vivo comes from cancer biology. Nigericin is considered to be a promising anticancer drug that demonstrates anticancer effects in various kinds of malignant tumors in vitro and in vivo [reviewed in 40]. Though the powerful anticancer effects of nigericin were highlighted in these studies, potential adverse effects of its application were also observed. The latter included teratogenic effects, insulin resistance, and eryptosis [56,57,58]. In this regard, an interval dosing strategy necessary and sufficient to achieve senolysis, contrary to constant application of the drug, might allow for the minimizing of unwanted adverse effects when applied in vivo.

To sum up, within the present study we proposed the novel approach for senolysis based on the disturbance of several homeostasis-maintaining systems and by applying uncovered nigericin as an effective senolytic compound. In addition, we verified the broad-spectrum senolytic activity of nigericin towards senescent cells and revealed molecular mechanisms underlying its senolytic action. Our data suggest that ionophore nigericin may offer a novel therapeutic strategy that potentially could be used against age-related diseases.

## 4. Materials and Methods

### 4.1. Cells Culture

END-MSCs, WJ-MSCs, DP-MSCs, and A549 were obtained from the Russian Collection of Cell Cultures (Institute of Cytology, Saint-Petersburg, Russia). The A549 line was authenticated by karyology, tumorigenicity, isoenzyme (LDH and G6PD) tests and STR analysis. Cells were cultured at 37 °C in a humidified incubator containing 5% CO_2_ in complete medium DMEM/F12 (Gibco BRL, Grand Island, NY, USA) supplemented with 10% FBS (HyClone, Logan, OH, USA), 1% penicillin-streptomycin (Gibco BRL, USA) and 1% glutamax (Gibco BRL, Grand Island, NY, USA). All cells were routinely tested for mycoplasma contamination using PCR. END-MSCs, WJ-MSCs, DP-MSCs of passages 7–10 were used.

### 4.2. Cells Treatment Conditions

Senescence models of END-MSCs, WJ-MSCs, DP-MSCs, and A549 were performed as described in detail in our previous study [23]. In brief, for oxidative stress-induced senescence, END-MSCs/WJ-MSCs were treated with 200 µM/100 µM H_2_O_2_ (Sigma-Aldrich, USA) for 1 h in serum-free media. For doxorubicine-induced senescence, DP-MSCs were treated with 1 μM of doxorubicine (Veropharm, Moscow, Russia) for 3 days. In each case, stress-induced END-MSCs/WJ-MSCs/DP-MSCs were considered senescent not earlier than 14 days after treatment. For etoposide-induced senescence, A549 were treated with 3 µM etoposide (Veropharm, Russia) for 3 days and analyzed not earlier than 7 days after senescence induction. For replicative senescence, END-MSCs cultured up to the 10th passage were identified as control cells and cultured later than 25th were identified as senescent ones.

All experimental treatments of cells were performed in complete culture media 3 days post-treatment cell were harvested by trypsinization and analyzed. The following previously established senolytics were used: 25 µM FOXO4-DRI (Pepmic, Suzhou, China), 100 µM quercetin (Sigma-Aldrich, St. Louis, MO, USA), 500 nM dasatinib (Sigma-Aldrich, USA) and 100 µM quercetin, 1 µM ouabain (Sigma-Aldrich, USA). Nigericin (Sigma-Aldrich, USA) was used in concentrations of 1 μM and 10 μM, as indicated in figures descriptions. Intracellular pH levels were modulated using 1 mM amiloride (Sigma-Aldrich, USA) while for mitochondrial membrane polarization modulation, 100 µM valinomycin (Sigma-Aldrich, USA) was applied, and autophagosomes turnover was blocked using 1 µM NH_4_Cl (Sigma-Aldrich, USA). To rescue cells from nigericin-induced death, 10 µM KCl (Panreac, Barcelona, Spain) or 50 µM Z-VAD-fmk (R&D Systems, Minneapolis, MN, USA) was used.

### 4.3. Flow Cytometry Analysis

Measurements of cell proliferation, viability, cell size, autofluorescence, membrane depolarization, intracellular pH, intracellular reactive oxygen species (ROS) levels, mitochondrial membrane potential, and apoptosis rates were carried out by flow cytometry. Flow cytometry was performed using the CytoFLEX (Beckman Coulter, Brea, CA, USA) and the obtained data were analyzed using CytExpert software version 2.0 (Beckman Coulter, Brea, CA, USA). Adherent cells were rinsed twice with PBS and harvested by trypsinization. Detached cells were pooled and resuspended in the fresh medium. In order to access cell viability, 0.1 μg/ml DAPI (Life Technologies, Carlsbad, CA, USA) was added to each sample just before analysis. DAPI-negative (living) cells were then counted and analyzed for autofluorescence to evaluate lipofuscin accumulation. The cell size was evaluated by cytometric forward light scattering. The loss of mitochondrial membrane potential was assessed using the ratiometric dye JC-1 (Invitrogen, Waltham, MA, USA), membrane depolarization was measured using a DiBAC4(3) fluorescent probe (Invitrogen, USA), to detect intracellular ROS cells were stained with 2’,7’-dichlorodihydrofluorescein diacetate (H_2_DCF-DA) dye (Invitrogen, USA), and intracellular pH levels were assessed using pH-indicator BCECF-AM (Invitrogen, USA), all dyes were used in accordance with the manufacture’s protocols. Apoptosis induction was verified using Annexin-V-APC (Invitrogen, USA) and DAPI co-staining following the manufacturer’s instructions.

### 4.4. Senescence-Associated Β-Galactosidase Staining

Senescence-associated β-galactosidase (SA-β-gal) staining was performed using a senescence β-galactosidase staining kit (Cell Signaling, Danvers, MA, USA) according to the manufacturer’s instructions. The quantitative analysis of images was produced with the application of the MatLab package, according to the algorithm described previously [59]. For each experimental point, 50 randomly selected cells were analyzed.

### 4.5. Western Blotting

Western blotting was performed as described previously [23]. SDS-PAGE electrophoresis, transfer to nitrocellulose membrane and immunoblotting with SuperSignal West Femto Maximum Sensitivity Substrate (Thermo Scientific, Waltham, MA, USA) detection were performed according to standard manufacturer’s protocols (Bio-Rad Laboratories, Hercules, CA, USA). Antibodies against the following proteins were used: glyceraldehyde-3-phosphate dehydrogenase (GAPDH) (clone 14C10), phospho-p53 (Ser15) (clone 16G8), p21 (clone 12D1), phospho-Rb (Ser807/811) (clone D20B12), phospho-ULK1 (Ser555) (clone D1H4), phospho-AMPKα (Thr172) (clone D4D6D), ATG9 (clone D4O9D), LC3A/B (clone D3U4C), and HRP-linked goat anti-rabbit IgG. Dilution rates were 1:1000 for all primary antibodies and 1:7000 for secondary ones. All antibodies were purchased from Cell Signaling, Danvers, MA, USA.

### 4.6. Enzyme Linked Immunosorbent Assay

The amounts of secreted IL-1β were measured in the conditioned media by the Human IL-1β ELISA Kit (Cytokine, Saint Petersburg, Russia) according to the manufacturer’s instructions. To determine the concentration of IL-1β in samples, GraphPad Prism 5 was used. Estimated amounts of IL-1β were normalized to total protein concentrations in the corresponding samples using the Bradford method.

### 4.7. Transcriptomic Analysis

Samples from our previously obtained RNA sequencing dataset GSE160702 were analyzed: samples GSM4877895—GSM4877898 and GSM4877907—GSM4877910 for control and senescent END-MSCs, respectively, were used. Reads processing, lightweight-mapping, and transcript abundances estimation were conducted using FastqPuri (1.0.7), BBtools (38.75), salmon (1.1.0) and tximeta (1.4.5), as described in detail in our recent studies [23,60]. Expression count matrices were summarized to a gene level and samples were filtered to contain genes having at least two estimated counts across half of the samples. Following differential expression testing, a log fold change estimation and GSEA-based functional annotation were performed using DESeq2 (1.26.0) and clusterProfiler (3.14.3) R packages as described in [23,25]. The Benjamini-Hochberg procedure was applied to correct for false positives. Full results of differential expression testing and functional annotation are presented in Appendix A, respectively.

### 4.8. Statistical Analysis

To get significance in the difference between two groups, a Welch’s *t*-test was applied. For multiple comparisons between groups, one- or two-way ANOVA with Tukey’s HSD was used. Unless otherwise indicated, all quantitative data are shown as mean ± s.d., and the asterisks indicate significant differences as follows: ns, not significant, * *p* < 0.05, ** *p* < 0.01, *** *p* < 0.005. Statistical analysis was performed using R software (4.1) (R Core Team, Vienna, Austria).

## 5. Conclusions

To summarize, within the present study we proposed a novel approach for MSCs senolysis based on the disturbance of the several homeostasis-maintaining systems. By applying this approach we uncovered nigericin as an effective senolytic compound. In addition, we verified the broad-spectrum senolytic activity of nigericin towards senescent cells and revealed molecular mechanisms underlying its senolytic action. Our data suggest that ionophore nigericin is an effective senolytic compound towards senescent MSCs and can be further tested as a new therapeutic strategy against age-related diseases.

## Figures and Tables

**Figure 1 ijms-23-14251-f001:**
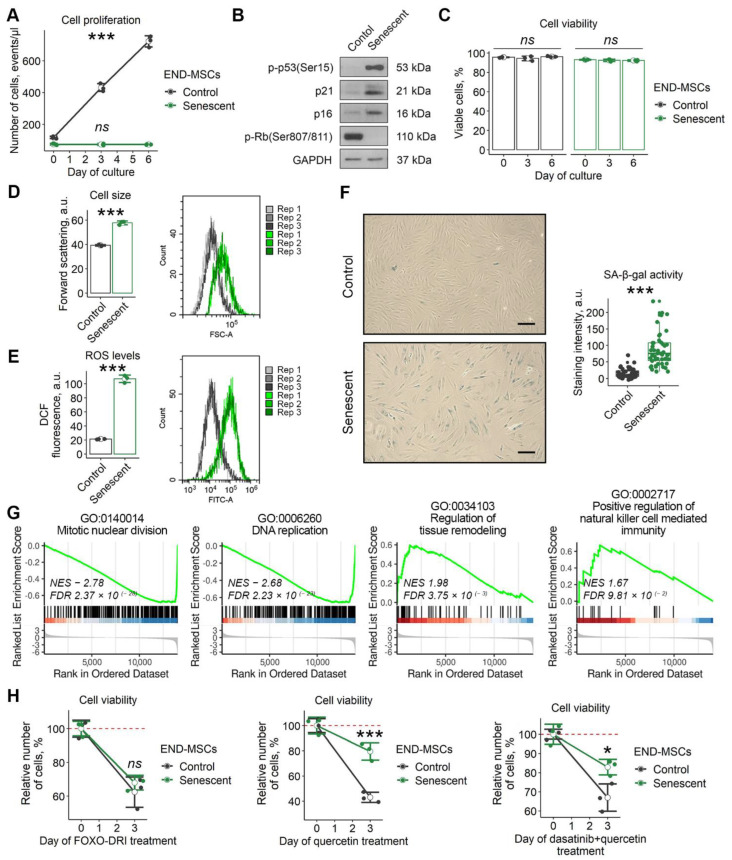
Senescent END-MSCs exhibit resistance to established senolytics treatment. The senescence state of END-MSCs was induced by oxidative stress (1 h, 200 μM H_2_O_2_) and verified by (**A**) proliferation loss and (**B**) activation of p53/p21/Rb and p16/Rb pathways mediating cell cycle arrest, (**C**) retained viability, (**D**) cell hypertrophy, (**E**) elevated intracellular ROS levels, (**F**) acquisition of senescence-associated activity β-galactosidase (SA-β-gal), (**G**) suppression of mitotic machinery, DNA replication and activation of tissue remodeling and pro-inflammatory gene expression profile compared to control cells (Appendix A). (**H**) Senescent END-MSCs display equivalent (25 μM FOXO4-DRI) or higher (100 μM quercetin, 500 nM dasatinib + 100 μM quercetin) resistance to death, data presented as relative cell viability (%) of control and senescent END-MSCs 3 days after treatment with each substance. Values presented are mean ± s.d. For multiple group comparisons at (**A**,**C**) one-way ANOVA was applied, *n* = 3, ns—not significant, *** *p* < 0.005. For pair comparisons at (**B**,**C,E**) Welch’s t-test was used, *n* = 3 for (**D**,**E**), *n* = 50 for (**F**), *** *p* < 0.005. For comparison of viability of control and senescent cell with treatment at (**H**) two-way ANOVA with Tukey’s HSD was applied, *n* = 3, ns—not significant, * *p* < 0.05, *** *p* < 0.005. For functional annotation of senescent vs. control END-MSCs transcriptomic profiles, comparison GSEA (Gene set enrichment analysis) in GO:BP terms was used, *n* = 4 for each group (Appendix A). Scale bars for images are 100 μm. Full-length blots are presented in Appendix A; GAPDH was used as loading control.

**Figure 2 ijms-23-14251-f002:**
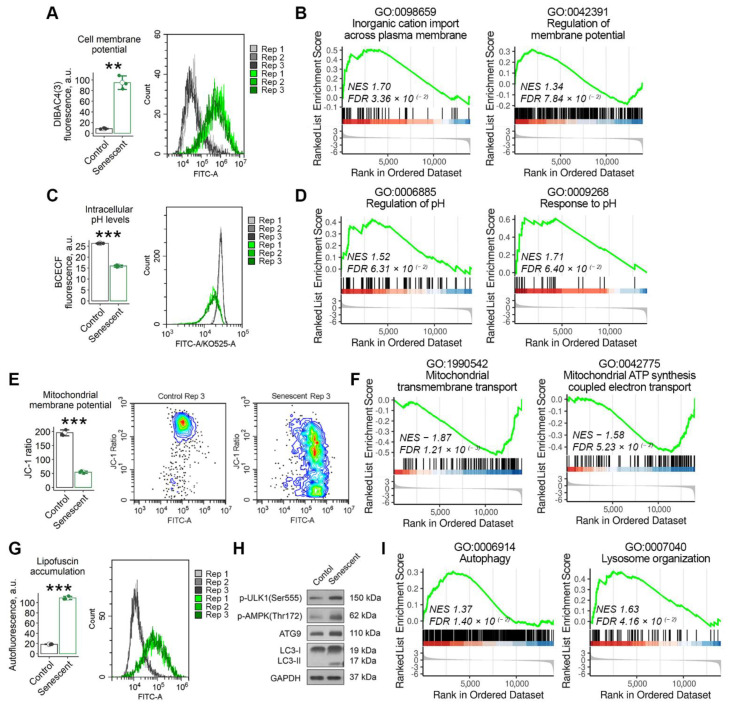
Senescent END-MSCs possess altered cell homeostasis. Compared to control END-MSCs, the senescent ones are characterized by (**A**,**B**) cell membrane depolarization, (**C**,**D**) lower cytoplasmic pH levels, (**E**,**F**) mitochondrial membrane depolarization, (**G**) accumulation of lipofuscin granules, and (**H**,**I**) the activation of autophagy and lysosome organization, including up-stream regulators (ULK1, AMPK) and down-stream mediators (ATG9, LC3) of autophagy. The observed differences between senescent and control END-MSCs were verified at the transcriptomic level using GSEA functional annotation in GO:BP terms, *n* = 4 for each group (Appendix A). Values presented are mean ± s.d. *n* = 3, ** *p* < 0.01, *** *p* < 0.005 using Welch’s *t*-test. Full-length blots are presented in Appendix A; GAPDH was used as loading control.

**Figure 3 ijms-23-14251-f003:**
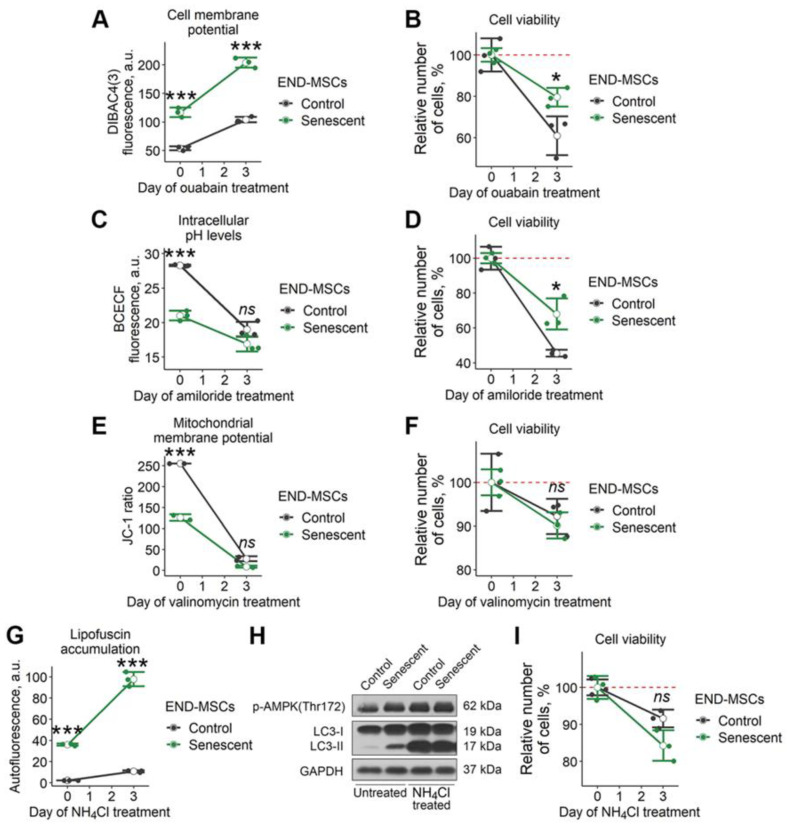
Senescent END-MSCs retain viability during modulation of separate cell homeostasis parameters. Senescent and control END-MSCs were treated with substances that modulate separate cell physiological parameters and were measured for viability after 3 days of exposure. (**A**,**B**) Cell membrane potential changes and viability under 1 μM ouabain treatment. (**C**,**D**) Intracellular pH levels and viability during 1 mM amiloride treatment. (**E**,**F**) Mitochondrial membrane polarization and viability during 100 μM valinomycin treatment. (**G**–**I**) lipofuscin accumulation, autophagy activity indicators and viability during 1 μM NH_4_Cl treatment. Cell viability data presented as relative cell viability (%) of control and senescent END-MSCs 3 days after treatment with each substance. Values presented are mean ± s.d. For comparison of parameters and viability of control and senescent cells with treatment, two-way ANOVA with Tukey’s HSD was applied, *n* = 3, ns—not significant, * *p* < 0.05, *** *p* < 0.005. Full-length blots are presented at Appendix A; GAPDH was used as loading control.

**Figure 4 ijms-23-14251-f004:**
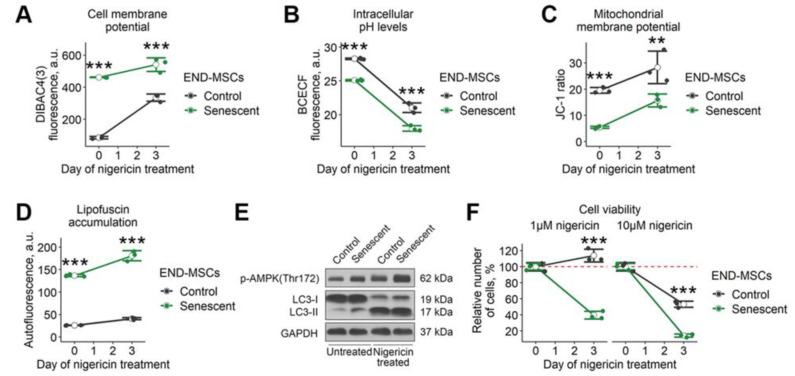
Nigericin treatment affects multiple cell homeostasis parameters and causes senolysis in END-MSCs. Nigericin treatment (10 μM) (**A**) diminished cell membrane potential, (**B**) lowered intracellular pH_i_ levels, (**C**) increased mitochondrial membrane polarization, (**D**) stimulated lipofuscin accumulation, and (**E**) modulated autophagy dynamics in control and senescent END-MSCs. (**F**) Nigericin treatment induced cell death more intensely in senescent END-MSCs compared to control ones at different concentrations, with cell viability data presented as relative cell viability (%) of control and senescent END-MSCs 3 days after treatment. Values presented are mean ± s.d. For comparison of parameters and viability of control and senescent cells with treatment, two-way ANOVA with Tukey’s HSD was applied, *n* = 3, ns—not significant, ** *p* < 0.01, *** *p* < 0.005. Full-length blots are presented at Appendix A; GAPDH was used as loading control.

**Figure 5 ijms-23-14251-f005:**
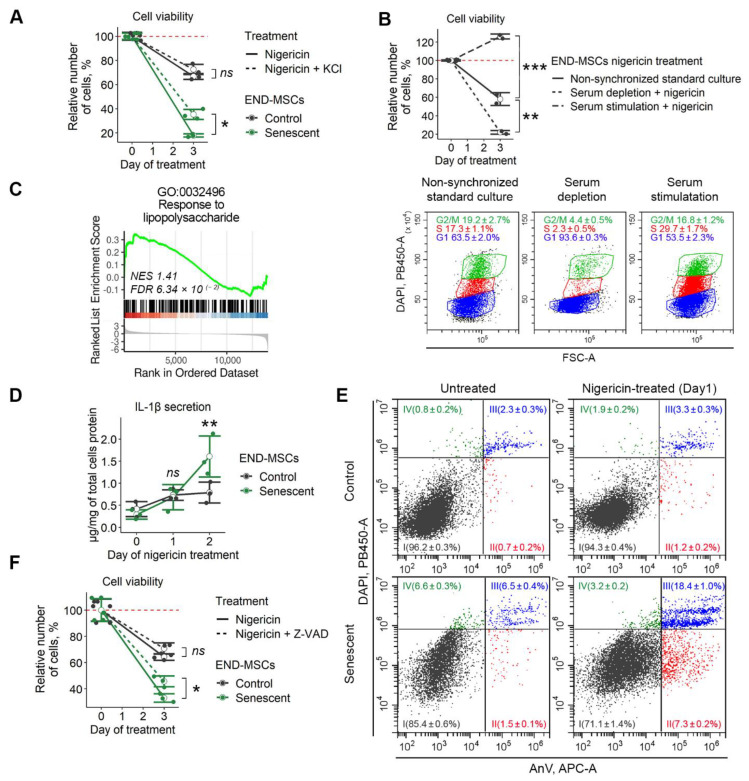
Nigericin treatment causes K+ efflux and initiates pyroptosis in senescent END-MSCs. Unless otherwise stated, nigericin was used at a concentration of 1 μM. (**A**) Nigericin-induced death of senescent END-MSCs was inhibited by supplementing culture media with 10 μM KCl. (**B**) Synchronization of control cells by serum depletion or subsequent serum level restoration modulated the toxic effect of 10 μM nigericin. (**C**) Differentially expressed genes between senescent and control END-MSCs are abundant in genes related to lipopolysaccharide-mediated pyroptotic response (Appendix A). (**D**) Nigericin treatment induced the pronounced secretion of IL-1β in senescent END-MSCs compared to control ones. (**E**) Annexin V/DAPI double staining reveals marked death induction of senescent END-MSCs compared to control cells. (**F**) Nigericin-induced death of senescent END-MSCs was inhibited by supplementing culture media with 50 μM Z-VAD-fmk pan-caspase inhibitor. Cell viability data presented as relative cell viability (%) of control and senescent END-MSCs 3 days after treatment. Values presented are mean ± s.d. ns—not significant, * *p* < 0.05, ** *p* < 0.01, *** *p* < 0.005, *n* = 3, one- or two-way ANOVA with Tukey’s HSD.

**Figure 6 ijms-23-14251-f006:**
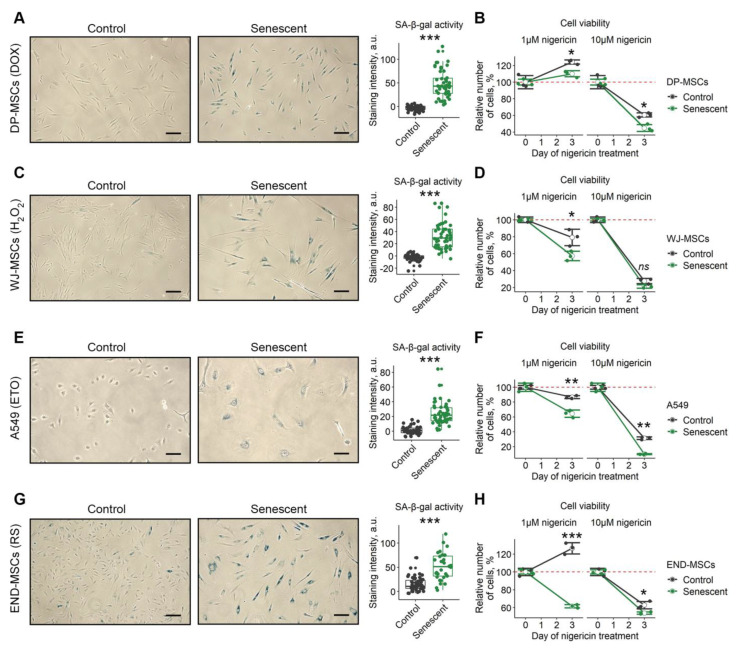
Nigericin can be used as a senolytic for cells of various origins. Four additional models of senescence of different cell lines was verified for acquisition of SA-β-gal activity and tested for the selective death of senescent cells: (**A**,**B**) control and doxorubicin-induced senescent (1 μM, DOX) DP-MSCs; (**C**,**D**) control and oxidative-stress induced senescent (100 μM, H_2_O_2_) WJ-MSCs; (**E**,**F**) control and etoposide-induced senescent (3 μM, ETO) A549 cells; (**G**,**H**) control and replicatively senescent (RS) END-MSCs. Cell viability data presented as relative cell viability (%) of control and senescent cells 3 days after treatment. Values presented are mean ± s.d. For pair comparisons at (**A**,**C**,**E**,**G**), Welch’s t-test was used, *n* = 50, *** *p* < 0.005. For comparison of viability of control and senescent cells with treatment at (**B**,**D**,**F**,**H**), two-way ANOVA with Tukey’s HSD was applied, *n* = 3, ns—not significant, * *p* < 0.05, ** *p* < 0.01, *** *p* < 0.005. Scale bars for images are 100 μm.

**Figure 7 ijms-23-14251-f007:**
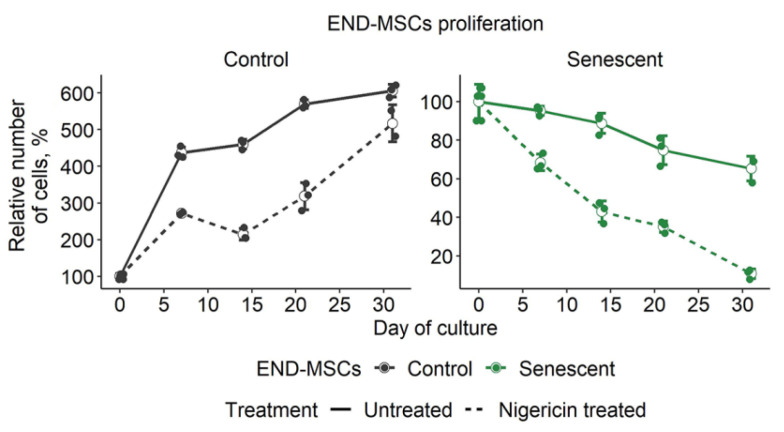
Repeated treatments with nigericin at low doses effectively induced senolysis in senescent END-MSCs and preserved the proliferation of control cells. Control and senescent END-MSCs were treated with 1 μM nigericin for 1 h every 7 days and was observed in culture for 30 days. Values presented are mean ± s.d.

## Data Availability

All data generated or analyzed during this study are included in the manuscript and supporting files. Sequencing data have been deposited in GEO under accession codes GSE160702. The custom code used in the study is available from the corresponding author upon request.

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
