# Peer review of "Targeting Multiple Homeostasis-Maintaining Systems by Ionophore Nigericin Is a Novel Approach for Senolysis"

_ijms, 2022, doi:10.3390/ijms232214251_

Round 1
Reviewer 1 Report
This manuscript is very interesting to propose the role of ionophore for control of senolysis. This approach is expected to be developed further. Relatively well written, but there are some points to be answered.
1. The major data are based on oxidative stress induced senescent cells. But there are huge differences between the SIPS cells and the replicative senescent cells. Therefore, it is necessary to show the data with the replicative senescent cells as far as possible.
2. In order to show the role of ionophore in senolysis by nigericin, it is necessary to show the nigericin sensitivity between young and old cells.
Since senolytics are supposed to eliminate the senescent cells, it is necessary to figure out the values of LD50 by nigericin in young and old cells.
3. Since the ionophore is assumed to play the major role, the comparison of transcriptional level or protein level of the ionophore is recommended.
Author Response
Reviewer 1.
We would like thank the Reviewer for the precious comments and suggestions.
- The major data are based on oxidative stress induced senescent cells. But there are huge differences between the SIPS cells and the replicative senescent cells. Therefore, it is necessary to show the data with the replicative senescent cells as far as possible.
Thanks for the notion. Indeed, most of the experiments in the study are preformed using oxidative stress induced senescent cells. However, to demonstrate broad-spectrum senolytic activity of nigericin, we performed the set of experiments using various forms of senescence, including the replicative one, and different types of cells (Fig. 6). As shown in Fig. 6G, nigericin demonstrates clear senolytic properties towards repicatively senescent END-MSCs similar to that towards stress-induced (oxidative stress) senescent cells.
- In order to show the role of ionophore in senolysis by nigericin, it is necessary to show the nigericin sensitivity between young and old cells. Since senolytics are supposed to eliminate the senescent cells, it is necessary to figure out the values of LD50 by nigericin in young and old cells.
Thanks for the comment. Different sensitivities of control and senescent cells towards nigericin are shown in Figures 4F, 6B,D,H,F. According to these results, nigericin preferentially kills senescent cells (of different origins and various senescence inducers) compared to normal cells (at concentration which kills nearly 50% of control cells). In terms of senolysis, exact LD50 values do not make much sense; far more important is to find the concentration of the drug that would have minimal effect on the viability of normal cells and at the same time would be effective in killing senescent ones. The later is typically achieved by minimizing the doze of the drug and performing interval dosing strategy. The results of such approach are shown in Fig. 7 – 1h pulse treatments with 1 μM nigericin repeated weekly allowed to eliminate 90 % of senescent cells without dramatic effects on the viability and proliferation of the control ones.
- Since the ionophore is assumed to play the major role, the comparison of transcriptional level or protein level of the ionophore is recommended.
Thanks for the suggestion. Ionophore nigericin is the lipid-soluble antibiotic that forms complexes with K+ at one side of the membrane and then migrates to the other side, where K+ is discharged and the anionic form is protonated. This compound is added exogenously and does not express in the cells, thus its levels cannot be assessed neither at mRNA nor at protein levels.
Reviewer 2 Report
Deryabin et al. demonstrated a novel method of developing senolytics by disrupting the intracellular ionic balance. Inducing senescent cells toward pyroptosis is a novel attempt at achieving senolysis, which certainly requires further validation by testing in aged animals. The study is novel and significant; however, I have specific concerns below:
Main:
1. The introduction is redundant. I suggest focusing on giving the rationale and hypothesis only.
2. It would be more convincing if the authors provided p16 data in fig.1B.
3. In figure 1H, to test if senescent END-MSCs are resistant to all senolytics, the authors should do dose titration curves and then compare EC50 of each drug
4. The authors did not describe the method being used for the cell viability assay
5. I suggest the authors provide EC50 for nigericin in Fig.4
6. The discussion is redundant. Please focus on your study's novelty, significance, impact, and limitation.
7. Pyroptosis will cause increased inflammation. The authors might add some discussion about whether nigericin-induced pyroptosis can cause complications and potentially spread senescence in older people. Also, the authors should mention the follow-up of the in vivo study to validate novel senolytics, nigericin.
Minor:
1. Please provide the concentrations of the senolytics in the figure legend for Fig.1H.
2. There are a few typos/grammars issues
Author Response
Reviewer 2.
We would like to thank the Reviewer for highlighting the strengths of our study as well as for pointing out several limitations.
Main:
- The introduction is redundant. I suggest focusing on giving the rationale and hypothesis only.
Thanks for the comment. We shortened the introduction to avoid redundancy. The second paragraph was removed from the text: “The effectiveness and specificity of senolytic compounds towards senescent cells largely depend on the possibility to uniquely identify these cells within tissues. Though senescent cells, in contrast to the cancer ones, do not bear any specific driver mutations that can easily distinguish them from normal cells, they demonstrate altered functioning of almost all intracellular systems [1, 16, 24]. The latter include morphologic features (increased size and granularity), disturbed functioning of mitochondria (altered mem-brane potential, increased reactive oxygen species production, disturbed oxidative phosphorylation) and lysosomes (senescence-associated beta-galactosidase (SA-β-gal) activity and lipofuscine accumulation), nuclear alterations (persistent DNA damage, loss of the lamin B1, specific chromatin remodeling patterns termed senescence-associated heterochromatin foci), etc. [1, 16, 25–30]. However, the precise epigenetic, transcriptomic, proteomic and metabolomic alterations that accompany senescence depend on cell types and inducing stimuli what makes it difficult to identify a universal set of features for senescent cells of different origins [31]. Additionally, senescence progression and de-velopment are governed by the signaling pathways that cannot be targeted safely, since in most cases they are crucial for the functioning of normal cells [31]. Together these issues challenge the development of a universal senolytic compound and force researchers either to design specific treatment schemes for already developed compounds or to look for novel approaches to elaborate new compounds with senolytic properties”.
- It would be more convincing if the authors provided p16 data in fig.1B.
Thanks for the notion. We added p16 data in the Fig. 1B and in the Supplementary Figure.
- In figure 1H, to test if senescent END-MSCs are resistant to all senolytics, the authors should do dose titration curves and then compare EC50 of each drug.
Thanks for the suggestion. In fact, we used approximately EC50 for each drug for the normal cells (around 50% of control cells remain viable after FOXO4-DRI, quercetin and dasatinib+quercetin). Importantly, the same concentrations of the drugs were less effective towards senescent cells, clearly demonstrating the absence of the senolytic action. We believe that the exact values of EC50 would not affect the conclusion regarding the absence of the senolytic action of the tested compounds. Of note, doses and treatment schemes applied for these compounds were similar to those in the original articles uncovering these senolytics.
- The authors did not describe the method being used for the cell viability assay
Thanks for the comment. The description of the method used to assess cell viability is provided in Materials and Methods section Flow cytometry (“In order to access cell viability, 0.1 μg/ml DAPI (Life Technologies, USA) was added to each sample just before analysis.”). It is the common approach to distinguish between intact live (DAPI-negative) and dead (DAPI-positive) cells with lost membrane integrity.
- I suggest the authors provide EC50 for nigericin in Fig.4
Thanks for the suggestion. In terms of senolysis, exact EC50 values do not make much sense; far more important is to find the concentration of a drug that would have minimal effect on the viability of normal cells and at the same time would be effective in killing senescent ones. As can be seen in Fig. 4F 1 µM kills around 50% of senescent cells, while 10 µM kills around 50 % of control cells. These results demonstrate that senescent cells are approximately 10-times more sensitive towards nigericin that control ones. Moreover, minimizing the doze of a drug and performing interval dosing strategy (1h pulse treatments with 1 μM nigericin repeated weekly) allowed to eliminate 90 % of senescent cells without dramatic effects on the proliferation of the control ones (Fig. 7).
- The discussion is redundant. Please focus on your study's novelty, significance, impact, and limitation.
Thanks for the comment. We shortened the discussion to avoid redundancy. The following paragraph was removed from the text: “The first-generation senolytics were aimed to disrupt senescent cell anti-apoptotic pathways (SCAPs) – specific pro-survival pathways that were found to be upregulated in senescent cells [32]. These approaches included the combined action of the Src kinase in-hibitor dasatinib and the flavonoid quercetin, inhibition of BCL-2 family members, inhi-bition of the peptidase USP7, inhibition of the HSP90 chaperones, activating caspases, and interruption of p53 interaction with the anti-apoptotic transcription factor FOXO4 [2, 4, 32, 42–47]. Most of the compounds targeting SCAPs were stated to have a broad-spectrum senolytic activity by killing senescent cells of various origins, including endothelial cells, fibroblasts, and various cancer cells [22, 32, 48]. However, there are some exceptions, for example navitoclax, A1331852, A1155463 (compounds that target BCL-2 family members), quercetin, FOXO4-DRI (small penetrating peptide interrupting FOXO4 and p53interaction), 17-DMAG (HSP90 inhibitor) do not display senolytic properties towards senescent preadipocytes and senescent MSCs from bone marrow [22, 42, 46, 49, 50]. Within the present study we also revealed the absence of senolytic activity of quercetin, combination of dasatinib plus quercetin and FOXO4-DRI towards senescent END-MSCs. Moreover, in our previous study we observed that different types of cells acquire either pro- or anti-apoptotic profile during senescence, what might underlie variations in sensitivity of these cells towards SCAPs-mediated senolysis [23]. Hence, our results together with the literary data demonstrate some limitations of the senolytic approaches based on SCAPs modulation.
Later on the class of senolytic drugs has substantially expanded and new approaches were proposed [16]. Most of these approaches considered functional or biochemical distinctions between normal and senescent cells for preferential senolysis [51–56]”.
- Pyroptosis will cause increased inflammation. The authors might add some discussion about whether nigericin-induced pyroptosis can cause complications and potentially spread senescence in older people. Also, the authors should mention the follow-up of the in vivo study to validate novel senolytics, nigericin.
Thanks for the suggestion. The respected Reviewer raised very interesting and important question regarding nigericin-induced inflammation that might affect neighboring cells and spread senescence. However, our data does not allow discussing this issue sufficiently, since the study did not focus on this topic and more research is needed to discuss it. We are planning to investigate the outcomes of nigericin-induced senolysis on the neighboring cells in the nearest future.
The possibility and limitations associated with application of nigericin in vivo are provided in the following paragraph of the discussion: “Even though senescent END-MSCs turned out to be far more sensitive towards nigericin, this compound affected the properties of the control cells. In fact, all senolytic strategies may elicit undesirable off-target effects, which can be limited by various dosing strategies [16]. According to the literary data, an interval dosing strategy might be rather effective, since senescent cells take 7 days or more to accumulate and develop a SASP at least in vitro, and might potentially take the same time to start re-accumulating in vivo [16, 75, 76]. Indeed, we were able to minimize negative effects of nigericin on the control cells preserving its maximal senolytic effect by performing three rounds of weekly pulse treatments. The possibility to apply nigericin in vivo comes from cancer biology. Nigericin is considered to be a promising anticancer drug that demonstrated anticancer effects in various kinds of malignant tumors in vitro and in vivo [reviewed in 61]. Though powerful anticancer effects of nigericin were highlighted in these studies, potential adverse effects of its application were also observed. The latter included teratogenic effects, insulin resistance and eryptosis [77–79]. In this regard, interval dosing strategy necessary and sufficient to achieve senolysis contrarily to constant application of the drug might allow minimizing unwanted adverse effects when applied in vivo.”
Minor:
- Please provide the concentrations of the senolytics in the figure legend for Fig.1H.
We added the missing concentrations of the drugs in the figure legends.
- There are a few typos/grammars issues
We corrected typos and grammar issues. All the changes made are marked within the text of the manuscript using the “Track Changes” function.
Round 2
Reviewer 1 Report
The basis of this manuscript is interesting and appropriate. Though replies of the authors are not satisfactory, but I think it is worth to publish as it is.